# A Country-Level Empirical Study on the Fiscal Effect of Elderly Population Health: The Mediating Role of Healthcare Resources

**DOI:** 10.3390/healthcare10010030

**Published:** 2021-12-24

**Authors:** Bo Tang, Zhi Li

**Affiliations:** School of Public Policy and Administration, Chongqing University, Chongqing 400044, China; hipytea@gmail.com

**Keywords:** population aging, health burden, fiscal deficit, healthcare resource, DALYs

## Abstract

Demographic shifts towards an aging population are becoming a significant fiscal challenge for governments. Previous research has explored the fiscal consequences of the expanding elderly population, but the impact on the elderly’s health quality is less mentioned. The balanced relationship between elderly population health and public finance is a major concern of the global political agenda on the aging society. This article used cross-country panel data from 2000 to 2019 to examine the fiscal effect of the elderly health burden and the mediating role of healthcare resources. The results are demonstrated: The elderly health burden has a negative impact on fiscal balance, especially in aged society and longevity countries. Moreover, the mediating effect of healthcare resources is significant, whereby various forms of healthcare resources such as funds, labor, and facilities all have significant effects. Thus, the conceptual framework of elderly population health, healthcare resources, and public finance is confirmed that the elderly health burden specifically leads to the growing consumption of healthcare resources, which reduces the fiscal balance. It is concluded that reducing the elderly health burden and improving healthcare resource efficiencies are two feasible strategies to enhance fiscal sustainability.

## 1. Introduction

Population aging is becoming more widespread and accelerated. The global population aged 60 years and over was only 205 million in 1950, increasing to 810 million in 2012, and it is estimated to reach two billion by 2050. By then, the proportions of people aged 60 and over will be 10% in Africa, 24% in Asia and Oceania, 25% in Latin America and the Caribbean, 27% in North America, and 34% in Europe [1]. On the other hand, the proportion of people aged 65 and over increased from 7% to 14% in France in the space of 116 years, compared to only 68 and 46 years in the United States and the United Kingdom, respectively. Surprisingly, it is expected to take only 26 and 21 years in China and Brazil, respectively [2]. Population aging is often associated with health and medical services and is a major driver of growth in social welfare spending [3]. Due to the natural situations of physiological decline and decreased resistance, the elderly inevitably behaves more demand in healthcare and medical services [4]. As known to all, pool health and disease burden can bring a multitude of unfavorable macroeconomic consequences [5]. Along with the rising population aging and healthcare expenditure, many governments have been under more and more fiscal pressure. Japan, one of the most aging countries in the world, has nearly tripled its share of spending on medical care, nursing care, healthcare, and other social security areas from 17.5% in 1990 to 32.7% in 2014 [6]; a similar situation as in Europe [7]. As can be seen, the health conditions of the increasing elderly population are expected to be a challenge to public finance, and thereby it should be attached importance to figure out the fiscal effects and mechanisms for policy-making.

The macroeconomic impacts of the demographic structure have been extensively studied in many empirical and theoretical literature [8]. The fiscal implications of population aging have been one main subject of considerable academic attention. Past research can be broadly divided into three categories. The first focuses on exploring whether population aging affects public finance. Hondroyiannis and Papapetrou [9] found the low fertility and high old-age dependency ratios that characterized the demographic structure of Greece between 1960 and 1995. The “double aging” trend found that population aging increases public debt and fiscal expenditure, leading to a deterioration in fiscal development. The second category focuses on predicting how population aging will affect public finance in the future. Demographic transition is associated with the future fiscal impact of population aging in terms of significant tax increases, social security increases, and welfare cuts [10]. The public debt ratio in Germany will continue to rise along with population aging, rapidly reaching a threshold of 100% [11]. Some argued that public finance needs to be estimated in advance, and forward-looking strategies need to be adopted to restructure fiscal revenue and expenditure and reduce the debt burden to effectively cope with the full impact of population aging [12]. The third category focuses on dissecting the underlying mechanisms of the fiscal effects already caused by population aging. The linkage mechanism has been identified between aging and social security from the trend of social security benefits continuously increasing with the proportion of the elderly population in the United States since the 1970s. Since a more significant proportion of the elderly population means that the proportion of older voters will also increase, and since older voters will tend to vote for motions to increase social security benefits, the trend of continuously rising social security benefit spending is difficult to reverse [13]. Empirical evidence from the static panel data of the United States and 12 eastern European has shown the dependency ratio and its significant negative relationship with the level of fiscal social security spending [14].

Furthermore, literature shows that health is productive to promote economic growth and human development via improving the efficiency of individual labor and expanding labor supply [15]. In contrast, disease and poor health mean sickness, absenteeism, and less productivity, which may fail to posse human capital accumulation [16]. It is implied that the disease burden of the population is possible to induce an overwhelming fiscal burden. However, the elderly population does not always create a burden. In terms of the healthy elderly population, they can also play a positive role in the labor market or household production [17]. It is evident that disease and poor health of elderly people need public finance to secure health care services, so the health status of the elderly population has far-reaching fiscal implications.

Most discussions and analyses in previous studies have referred to the impact of population aging and population health, respectively, on public finance, but they have paid less attention to the fiscal effects of the elderly health. In fact, elderly people are not the determinant of fiscal deficits. Poor health and disease burden ought to be the real reason for putting governments immersed in debt. With these facts in mind, we find this is a significant gap in the literature and a lack of empirical evidence. It is also practical significant given that even advanced countries are still struggling to find money to pay for the health care costs for the rising elderly population [18].

The purpose of this study is to empirically identify the fiscal effect and mechanisms of the elderly health burden using the panel data from 45 countries from 2000 to 2019. The remainder of this paper is organized as follows: Section 2 provides a theoretical framework; Section 3 introduces the empirical strategy and study design; Section 4 reports the results; Section 5 presents associated analysis and discussion; Section 6 proposes the conclusions and policy suggestions.

## 2. Theoretical Framework

### 2.1. Basic Model

This study is based on a two-period intergenerational overlap model, assuming that the population in society has only two life cycles, young and old, representing the young-age working population and the old-age retired population, respectively, according to the general model [19,20]. Thus, *N**_t_* denotes the number of young people surviving in period *t*, *N**_t−_*_1_ denotes the number of old people surviving in period *t*, and the total population *P**_t_* is calculated by the following formula:(1)$$P_{t}=N_{t}+N_{t-1}$$

If the population growth rate is *n*, then the relationship between the number of young people and the number of old people in period *t* can be written as:(2)$$N_{t}=\left(1+n_{t-1}\right)\cdot N_{t-1}$$

Therefore, let $\alpha_{t}$ denote the proportion of the elderly population in period *t*, which can be expressed as:(3)$$\alpha_{t}=\frac{N_{t-1}}{P_{t}}=\frac{N_{t-1}}{N_{t-1}\cdot \left(2+n_{t-1}\right)}=\frac{1}{2+n_{t-1}}$$

Meanwhile, Equation (3) can be converted to:(4)$$n_{t-1}=\frac{1}{\alpha_{t}}-2$$

A balanced fiscal system is assumed, in which the fiscal revenue generated by the young-age working population exactly offsets the fiscal expenses used by the old-age retired population. With the fiscal revenue of each young person denoted as *T* and the fiscal expenses of each elderly person denoted as *S*, the fiscal revenue generated by the young population in initial period 0 fully offsets the fiscal expenses of the elderly population in the previous period, which can be expressed as follows:(5)$$N_{-1}\cdot S=N_{0}\cdot T$$

According to Equation (2), Equation (5) can be transformed into:(6)$$S=\left(1+n_{-1}\right)\cdot T$$

According to Equation (4), Equation (6) can be transformed into:(7)$$S=\left(\frac{1}{\alpha_{0}}-1\right)\cdot T$$

Thus, $\alpha_{0}$ denotes the share of the elderly population in base period 0. Let *D* denote the fiscal balance. Then, *D*_0_ denotes the fiscal balance in period 0, where its value is zero due to the revenue balancing the expenses. *D*_1_ denotes the fiscal balance in period 1, which can be expressed as the difference between the fiscal revenue generated by the young population in period 1 and the fiscal expenses spent by the elderly population in period 0.
(8)$$D_{1}=N_{1}\cdot T-N_{0}\cdot S$$

According to Equations (2) and (6), Equation (8) can be expressed as:(9)$$D_{1}=N_{0}\cdot \left(1+n_{0}\right)\cdot T-N_{0}\cdot \left(1+n_{-1}\right)\cdot T$$

According to Equation (4), Equation (9) can then be expressed as:(10)$$D_{1}=N_{0}\cdot \left[1+\left(\frac{1}{\alpha_{1}}-2\right)\right]\cdot T-N_{0}\cdot \left[1+\left(\frac{1}{\alpha_{0}}-2\right)\right]\cdot T=N_{0}\cdot T\cdot \left(\frac{1}{\alpha_{1}}-\frac{1}{\alpha_{0}}\right)$$

Assuming that the health burden per capita of all society, denoted *h*, is constant, Equation (10) can then be transformed into:(11)$$D_{1}=N_{0}\cdot T\cdot \left(\frac{1}{\alpha_{1}}-\frac{1}{\alpha_{0}}\right)=N_{0}\cdot T\cdot h\cdot \left(\frac{1}{\alpha_{1}h}-\frac{1}{\alpha_{0}h}\right)$$

In Equation (11), since *N*_0_, *T*, α, *h* > 0, the sign of *D*_1_ depends mainly on the difference between $\alpha_{0}h$ and $\alpha_{1}h$. Therefore, $\alpha_{0}h$ denotes the elderly health burden in base period 0. When $\alpha_{0}h<\alpha_{1}h$, indicating a rise in the elderly health burden, the fiscal balance *D*_1_ is negative, denoting a fiscal deficit; when $\alpha_{0}h>\alpha_{1}h$, indicating a decline in the elderly health burden, the fiscal balance *D*_1_ is positive, denoting a fiscal surplus; when $\alpha_{0}h=\alpha_{1}h$, the fiscal balance *D*_1_ is zero, indicating that the fiscal balance is maintained. Above all, it can be seen that the elderly health burden is inversely related to the fiscal balance.

### 2.2. Mechanism Analysis

Considering the relationship between the elderly population and public finance, there is a widespread belief that a growing elderly population will drive up the consumption of healthcare resources and cause an ever-widening federal deficit [21]. Health demand is a continuous social demand, which is an important factor in promoting fiscal sustainability and economic development. Healthcare resources are the general term of social resources consumed in the process of providing healthcare services, which is one measure of the capacity of the healthcare system [22]. According to the literature, there are generally the resources of funds, labor, and facility invested in the healthcare domain. First, one of them, the elderly health burden requires spending health funds and thereby reducing fiscal balance. It was found that health care costs were significantly higher in the older age group than in the younger age group from the evidence of Australia [23]. In terms of hospitalization costs, the frequency and cost of hospitalization were significantly higher in the elderly group than in other age groups [24]. There was a positive effect of the elderly population proportion on the fiscal deficit because population aging leads to a dramatic increase in medical costs and puts healthcare insurance funds under great pressure. On the other hand, the human resources for health are central to overcoming health crises in some pool countries and for building sustainable health systems in all world [25]. According to OECD countries, one in every ten jobs was in health or social care in 2019, which have employed the hugest workforce in history [26]. In addition, more health facilities have increased fiscal expenditure and thereby make a reduction of financial balance [27]. In summary, the elderly health burden causes more consumption of healthcare resources, which have an adverse impact on public finances. Therein, various forms of resources such as funds, labor, and facilities all have reduced the fiscal balance. Thus, the theoretical mechanism is characterized in Figure 1.

## 3. Materials and Methods

### 3.1. Empirical Strategies

First, the baseline model in Equation (12) was established on the basis of the analysis above, where *Balance* represents the fiscal balance, *Burden* represents the elderly health burden, and *Z* represents a set of control variables that affect the fiscal balance. Meanwhile, γ and δ are the coefficient terms of the regression variables, respectively, α is the intercept term, and ε represents the residual term.
(12)Balance=α+γBurden+δZ+ε

Coefficient estimation may be biased due to the fact that the heterogeneity of the sample and the disturbance brought about by the time series are not taken into account in the baseline model; hence, two-way fixed effects regression (2FE) was further constructed in Equation (13), where *u* and *y* represent the geographic region and the time of year in which the country is located, respectively, and the subscripts *i* and *t* denote the specific geographic region and year.
(13)$$Balance_{i,t}=\alpha +\gamma_{0}Burden_{i,t}+\delta Z_{i,t}+u_{i}+y_{t}+\varepsilon_{i,t}$$

Lastly, to test the mechanism of the elderly burden and the fiscal balance, the mediating effect models in Equations (14) and (15) were constructed, where *Mediation* denotes the mediating variable, and the remaining variables have the same meaning as above. Referring to Baron and Kenny’s method of mediating effects analysis [28], if the coefficients of the core explanatory variables in the main regression $\gamma_{0}$ are significant, then the coefficients are estimated from the mediating effects of $\gamma_{1}$ and  θ, where $\gamma_{1}$ represents the effect of the elderly health burden on the mediating variable, and θ then the effect of the mediating variable on the fiscal balance. If both coefficients are significant, this indicates that there is a mediating effect, whereby the elderly health burden can affect the fiscal balance through the mediating variable; if at least one of them is not significant, Sobel testing is needed to determine whether a mediating effect exists.
(14)$$Mediation_{i,t}=\alpha +\gamma_{1}Burden_{i,t}+\delta Z_{i,t}+u_{i}+y_{t}+\varepsilon_{i,t}$$
(15)$$Balance_{i,t}=\alpha +\gamma_{2}Burden_{i,t}+\theta Mediation_{i,t}+\delta Z_{i,t}+u_{i}+y_{t}+\varepsilon_{i,t}$$

### 3.2. Variable Descriptions

The dependent variable was Fiscal Balance, which was measured as a function of the ratio of the net operating balance to GDP (*Balance*). The net operating balance was calculated simply by the difference between public finance revenue and expense. A positive value of this indicator denotes a fiscal surplus. Conversely, a negative value of this indicator denotes a fiscal deficit. In addition, the ratio of net borrowing and lending to GDP (*Finance*) was used as an alternative indicator for robustness testing. The data of the net operating balance and net lending and borrowing were obtained from the Government Finance Statistics (GFS) published by the International Monetary Fund.

The independent variable was elderly health burden. There are many indicators to measure population aging, including the proportion of the elderly population, mortality rate, life expectancy, and life/course ratio [29]. In this paper, we adopted disability-adjusted life years (DALYs) as the measurable indicator of the health burden [30]. Thus, the elderly health burden was measured as a function of the disease burden rate of the elderly population (*Burden*), which is the DALY ratio of the population aged 65 and above to that of the total population. According to the World Health Organization (WHO), DALYs are the total number of healthy life years lost from onset to death, including both years of life lost due to premature death and years of healthy life lost due to disease-induced disability. In addition, the old-age dependency ratio (*Aging*), which is the ratio of the population aged 65 and over to the working-age population, was used as an alternative for robustness testing. The data required for the calculations were taken from the Global Health Data Exchange (GHDx) database.

Healthcare resources represented the mediating variable in this study, and indicators were set to measure this from the perspective of resources such as healthcare expenditure [31,32,33], human resources [34], and hospital beds [35], using three indicators, namely, healthcare expenditure rate (*THE*), physician ratio (*PHY*), and hospital bed ratio (*BED*), to represent the financial resources, labor resources, and facility resources invested in the healthcare domain, respectively, as a comprehensive measure of healthcare investment. The data of the mediating variable indicators were obtained from the World Health Statistics (WHS) database established by the World Health Organization.

With reference to control variables of previous studies, the factors affecting the fiscal balance were controlled from multiple perspectives, such as social, economic, and environmental [36,37,38]: (1) Urbanization (*URB*), measured by the ratio of the urban population to the total population of the country; (2) openness (*TRD*), measured by the ratio of the total import and export trade to the country’s GDP; (3) foreign direct investment (*FDI*), measured by the ratio of net foreign direct investment to the country’s GDP; (4) forestation (*FOR*), measured as the ratio of forest area to the country’s total land area; (5) growth (*PRG*), measured by the annual growth ratio of GDP per capita. The data for the control variables were obtained from the World Development Indicators (WDI) database published by the World Bank.

In all, Table 1 reports the selection and description of variables, indicators, and data sources.

### 3.3. Data Explanation

This study selected panel data of 45 countries for the period of 2000–2019 as the sample for analysis; in the process of data collation, the data completeness of the indicators for the explanatory and core explanatory variables was the priority criterion, and those samples with abnormal indicators and serious missing data were excluded. Then, interpolation was used to fill in the sporadic missing data for individual years. The statistical characteristics of the main variables are reported in Table 2.

## 4. Results

### 4.1. Basic Regression

During the basic regression analysis, changes in the relationship between the elderly health burden and the fiscal balance were observed by gradually adding control variables, and the results are reported in Table 3. Column 1 shows that the estimated coefficient of the elderly health burden without the inclusion of any control variables was −0.065, which had statistical significance at the 1% level. The results from Columns 2 to 6 show that the estimated coefficients of the elderly health burden ranging from −0.128 to −0.114 did not change much when control variables were gradually added, all of which were significantly negative at the 1% statistical level. It is indicated that the elderly health burden had a significant dampening effect on the fiscal balance and that countries or regions with a higher elderly health burden had a weaker fiscal balance. Considering the control variables, the results from Column 6 show that the estimated coefficients of all control variables were significant at the 1% or 5% statistical level. Among them, urbanization, openness, forestation, and GDP per capita growth had a significantly positive effect on the fiscal balance, indicating that a higher urbanization rate, more open trade, a greener environment, and faster GDP per capita growth lead to a better impact on public finance. In contrast, the effect of foreign direct investment on the fiscal balance was significantly negative at the 5% statistical level, indicating that a higher share of foreign investment leads to a lower fiscal balance. In addition, after controlling for the time and geographic effects of the sample countries, the effect of the elderly health burden on the fiscal balance remained significantly negative.

### 4.2. Endogeneity Concerns

To control for endogeneity issues of the model, this paper used lagged term as the instrumental variable for further dynamic regression analysis by system generalized method of moments (GMM) [39]. Table 4 shows the regression results of the system GMM estimation. Column 1 shows that the estimated coefficient of the first-order lagged term of fiscal balance is significantly positive at the 1% significant level, indicating that the current period of fiscal balance is positively correlated with the previous period. Moreover, the regression coefficient of the elderly health burden was significant at the 5% level, which indicated a negative-driving effect on fiscal balance. In addition, to avoid measurement error, we replaced the variable measurements to re-run the regressions by taking the ratio of net borrowing and lending to GDP (*Finance*) and the old-age dependency ratio (*Aging*) as an alternative. As the results are shown in Columns 2 to 4, the core explanatory variable remained significant except in Column 4. Finally, according to model specification tests, it is also demonstrated that moment conditions employed in the system GMM are appropriate—errors are uncorrelated of order 2 (AB test *p*-value > 0.1), and there are no overidentifying restrictions (Hansen test *p*-value > 0.1), thus indicating that the selection of instrumental variables and the setting of the model are reasonable.

### 4.3. Robustness Tests

The results of various robustness tests are reported in Table 5. First, given the cyclical nature of the fiscal budget and the possibility of inter-year effects on the fiscal balance, it was necessary to examine whether the current period’s level of the elderly health burden had an effect on the next period’s fiscal balance (*Balance_t+_*_1_). The results of Column 1 show that the regression coefficient of the elderly health burden was significantly negative at the 1% statistical level, consistent with the conclusion above.

Second, we replaced the OLS regression model in the basic regression with a logistic regression model, redefining the explained variable fiscal balance as a dummy variable (*Balance_dummy*), whereby a value of one was assigned for a fiscal surplus, and the value of zero was assigned for a fiscal deficit. According to the logistic regression results in Column 2, the findings did not change substantially from the conclusions above.

Third, selecting the restricted samples. In order to exclude the influence of sample outliers on the research findings, we conducted robustness tests by adjusting the sample groupings. On the one hand, the public finance situation was not only affected by the domestic economy but was subjected to shocks from global macroeconomic events. Thus, we retested the regression results after excluding the sample data during the financial crisis of 2008–2010, as shown in Column 3. On the other hand, sample data with too extreme fiscal surplus or deficit rates were excluded. The regression results are reported in Column 4 using the sample data with a fiscal balance rate in the range of ±15%. The results show that the regression coefficients of the elderly health burden were both significantly negative at the 1% statistical level, indicating that an increase in elderly health burden still significantly reduces the fiscal balance, posing a negative impact. The results remained robust.

### 4.4. Heterogeneity Analysis

Considering that country heterogeneity possibly affects the basic result, we have performed a heterogeneity analysis from grouping by OECD member, aged society, and life expectancy. The results are reported in Table 6. First, from the subgroup results of whether OECD members in Columns 1 and 2, we can see that the elderly health burden has a significantly negative effect on fiscal balance at the 1% level, in both groups. Second, according to the United Nations, an “aged society” is defined as a country whose share of people aged 65 years or more has reached 14% of the total population. Thus, we classified the countries with the elderly population proportion less than 14% as being in the Pre-aged group, and more than 14%, the Aged group [40]. From the results presented in Columns 3 and 4, it is clear that the elderly health burden has a significant negative fiscal effect at the 1% level in aged society countries, but not in pre-aged society. Third, according to the rank of the sample countries by the mean value of life expectancy from 2000 to 2019, the median is 79.41 years old. Countries with an average life expectancy below 79.41 years old are classified as being in the Normal group, and countries with an average life expectancy above 79.41 years old are classified as being in the Longevity group. The results presented in Columns 5 and 6 show that the elderly health burden has negatively affected the fiscal balance in the Longevity group at the 1% level, but it is not significant in the Normal group. Overall, the reason for the significant fiscal impact of the elderly health burden in aged society and longevity countries may be due to the deep population aging and high life expectancy, which means a larger proportion of the elderly population over 75 or 80 years old, who are likely to incur greater costs in dying [41]. This is because the two years before dying is a period of high healthcare costs for the elderly [42,43], with the greater probability of higher spending on healthcare as death draws nearer [44].

### 4.5. Mediating Effect

Further, to recognize the mechanism of healthcare resources, the mediating effect was further analyzed. According to the research design, healthcare resources were measured in various forms, including funds, labor, and facilities. They were tested one-by-one in accordance with the model and analysis steps of the mediating effects to verify the existence of the three transmission mechanisms, and the results are reported in Table 7.

First, in terms of healthcare funds, the results in Columns 1 and 2 show that the regression coefficients of the elderly health burden on healthcare funds, and of healthcare funds on the fiscal balance, were statistically significant at the 1% level. Thus, it is straightforward to conclude that a mediating effect exists without the need for Sobel testing. Column 1 shows that the effect of the elderly health burden on health funds was significantly positive, while Column 2 shows that the effect of healthcare funds on the fiscal balance was significantly negative, indicating that a rise in the elderly health burden increases the health expenditure and, thus, reduces the fiscal balance.

Second, in terms of healthcare labor, the estimates in Columns 3 and 4 show that the two-stage regression coefficients of the elderly health burden on healthcare labor and of healthcare labor on the fiscal balance were statistically significant at the 1% level. Thus, a mediating effect existed. Column 3 reveals a significant positive effect of the elderly health burden on healthcare labor, while Column 4 reveals a significant negative effect of healthcare labor on the fiscal balance, demonstrating that an increase in the elderly health burden necessitates increased healthcare labor, thus reducing the fiscal balance.

Third, in terms of healthcare facilities, the regression coefficients of the elderly health burden on healthcare facilities and of healthcare facilities on the fiscal balance in Columns 5 and 6 were statistically significant at the 1% level; hence, a mediating effect existed. Column 5 shows a significant positive effect of the elderly health burden on healthcare facilities, while Column 6 shows a significant negative effect of healthcare facilities on the fiscal balance, indicating that an increase in the elderly health burden necessitates increased healthcare facilities for support, which decreases the fiscal balance.

## 5. Discussions

This paper empirically investigated the fiscal effect of elderly population health and its mediation mechanism through cross-country panel data, and the main findings were as follows: (1) The regression results of the fixed effects model showed that the impact of the elderly health burden on the fiscal balance was significantly negative, suggesting that an increase in the elderly health burden adversely affects the fiscal balance; (2) the heterogeneity effects have been identified in different countries, the elderly health burden performed a significantly negative influence on the fiscal balance in aged society and longevity countries; and (3) the results of the mediation effect tests showed that a rising elderly health burden leads to an increase in healthcare resources, which in turn has a negative impact on the fiscal balance.

Taken together, such results keep consistent with most of the previous studies [45,46]. It is explainable that when the disease burden of the elderly population in society rises, the social demand for healthcare increases. Healthcare is the main aspect of public services, which is related to the livelihood of the country; thus, the government has a responsibility to provide basic healthcare services. Accordingly, the budget structure is adjusted to increase healthcare resources, forming a crowding-out effect [47]. This can also be analyzed from the perspective of supply and demand, whereby an increased elderly health burden leads to a rise in a society’s healthcare on the demand side; hence, the government has to increase the fiscal investment to meet the growing health demand on the supply side [48]. This analysis confirms that an increase in the elderly health burden leads to an increase in healthcare resource consumption, which in turn leads to a decline in the fiscal balance.

The limitation of this study is mainly that it lacks a detailed analysis of healthcare resources. Actually, healthcare resources encompass a wide range of components, and in this paper, only three terms of healthcare resources were discussed, which may not be fully representative, so the final research conclusions have less completeness and guidance for fiscal policies on population aging. Future research is expected to cover this limitation and strengthen the conclusions of this study.

## 6. Conclusions

International experience shows that rising healthcare expenditure can be a dangerous shock to fiscal sustainability, whereby higher welfare is often associated with higher debt [49]. Previous studies have focused on the impact of the elderly population quantity on public finance, but the elderly health quality is rarely considered. This study, from a new perspective of disease burden, has demonstrated that the elderly health burden, which could be a new indicator of population aging, has a negative effect on fiscal balance, and the mediation mechanism of healthcare resources on the fiscal effect of elderly health burden has been confirmed. Therefore, it is beneficial for governmental fiscal balance to reduce the disease burden of the elderly population and improve the utilization efficiency of healthcare resources.

Based on the above findings, some policy recommendations could be considered. The first one is to consider “aging and health” as the top priority in fiscal policies. It is necessary to encourage the innovation of health financial products such as pension funds, chronic disease medical funds, and long-term care insurance to cover health shocks and risks. The second is to improve the efficiency of healthcare resources to respond to population aging. “Value-based healthcare” [50] is supposed to be effective for elderly healthcare to optimize higher health outcomes and lower health costs. The third is to promote to elderly people a sense of “active health” in order to keep healthy and reduce disease burden. Health literacy for the elderly should be improved in order to inform them on how to conduct health management correctly and actively.

## Figures and Tables

**Figure 1 healthcare-10-00030-f001:**
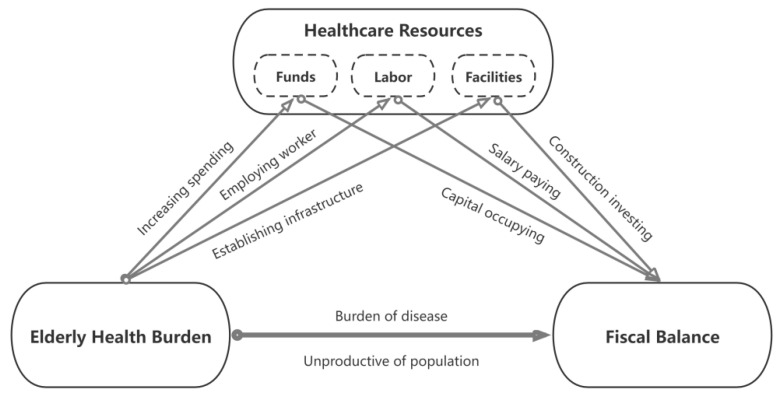
Theoretical framework chart.

**Table 1 healthcare-10-00030-t001:** Selection and description of variables, indicators, and data sources.

Variable	Indicator	Sign	Definition	Data Source
Fiscal balance	Budget balance rate	Balance	Net operating balance/GDP	GFS database
	Financial balance rate	Finance	Net lending or borrowing/GDP	GFS database
Elderly health burden	Elderly disease burden rate	Burden	DALYs aged 65+ years/DALYs all ages	GHDx database
	Aged dependency ratio	Aging	Population aged 65+ years/population aged 15–64 years	WDI database
Healthcare resource	Health expenditure rate	THE	Current health expenditure/GDP	WHS database
	Physician ratio	PHY	Physicians per 1000 people	WHS database
	Hospital bed ratio	BED	Hospital beds per 1000 people	WHS database
Urbanization	Urbanization rate	URB	Urban population/total population	WDI database
Openness	Trade share	TRD	Total trade volume/GDP	WDI database
Foreign direct investment	Foreign investment share	FDI	Foreign direct investment net inflow/GDP	WDI database
Forestation	Forest coverage rate	FOR	Forest area/land area	WDI database
Growth	Economic growth rate	PRG	GDP per capita growth rate (%)	WDI database

**Table 2 healthcare-10-00030-t002:** Descriptive statistics of the indicators.

Sign	Obs	Mean	SD	Min.	Max.
Balance	900	−0.471	4.328	−30.576	19.979
Finance	900	−1.730	4.080	−32.066	18.633
Burden	900	41.044	9.995	7.993	61.916
Aging	900	22.080	6.992	7.149	47.122
THE	900	7.856	2.343	1.909	17.709
PHY	900	2.856	1.072	0.130	6.630
BED	900	5.015	2.548	0.125	14.690
URB	900	73.194	13.99	31.386	98.041
TRD	900	97.419	60.005	19.798	408.362
FDI	900	8.413	30.485	−58.323	449.083
FOR	900	34.443	18.370	0.298	73.736
PRG	900	2.222	3.163	−14.269	23.986

**Table 3 healthcare-10-00030-t003:** OLS regression results.

	(1)	(2)	(3)	(4)	(5)	(6)
Burden	−0.065 ***	−0.128 ***	−0.117 ***	−0.117 ***	−0.128 ***	−0.114 ***
	(−5.49)	(−5.30)	(−4.80)	(−4.81)	(−5.24)	(−4.96)
URB		0.047 ***	0.046 ***	0.047 ***	0.056 ***	0.069 ***
		(4.67)	(4.73)	(4.77)	(5.50)	(6.72)
TRD			0.012 ***	0.013 ***	0.013 ***	0.011 ***
			(5.45)	(5.92)	(6.40)	(5.43)
FDI				−0.010 ***	−0.008 ***	−0.006 **
				(−3.29)	(−2.75)	(−2.29)
FOR					0.035 ***	0.032 ***
					(4.65)	(4.48)
PRG						0.302 ***
						(4.94)
Constant	2.182 ***	3.036 ***	2.052 **	1.963 *	0.017	−2.501 **
	(4.35)	(2.86)	(2.03)	(1.94)	(0.02)	(−2.09)
Year fixed	No	Yes	Yes	Yes	Yes	Yes
Region fixed	No	Yes	Yes	Yes	Yes	Yes
*R* ^2^	0.022	0.185	0.205	0.209	0.224	0.252
Obs	900	900	900	900	900	900

Note: *** *p* < 0.01, ** *p* < 0.05, and * *p* < 0.1; standard errors in parentheses.

**Table 4 healthcare-10-00030-t004:** System GMM estimation results.

Variable	(1)	(2)	(3)	(4)
	Balance	Balance	Finance	Finance
L.Balance	0.845 ***	0.897 ***		
	(0.096)	(0.107)		
L.Finance			0.826 ***	0.855 ***
			(0.091)	(0.095)
Burden	−1.113 **		−1.052 *	
	(0.521)		(0.522)	
Aging		−5.625 *		−5.696
		(3.030)		(3.420)
URB	−0.021	−0.035	−0.015	−0.031
	(0.030)	(0.040)	(0.029)	(0.039)
TRD	0.005 *	0.006	0.007	0.009
	(0.003)	(0.005)	(0.004)	(0.006)
FDI	−0.012	−0.019	−0.012	−0.018
	(0.011)	(0.012)	(0.012)	(0.012)
FOR	0.039	0.055	0.022	0.042
	(0.055)	(0.058)	(0.053)	(0.061)
PRG	−0.209	−0.310 *	−0.306	−0.389 **
	(0.207)	(0.160)	(0.263)	(0.181)
AR(1) Test *p*-value	0.003	0.004	0.002	0.004
AR(2) Test *p*-value	0.292	0.848	0.370	0.974
Hansen Test *p*-value	0.485	0.154	0.672	0.292
Obs	765	765	765	765

Note: *** *p* < 0.01, ** *p* < 0.05, and * *p* < 0.1; standard errors in parentheses.

**Table 5 healthcare-10-00030-t005:** Robustness test results.

Variable	(1)	(2)	(3)	(4)
	Balance_t+1_	Balance_dummy	Balance	Balance
Burden	−0.105 ***	−0.052 ***	−0.120 ***	−0.124 ***
	(0.023)	(0.015)	(0.023)	(0.021)
URB	0.065 ***	0.028 ***	0.070 ***	0.056 ***
	(0.010)	(0.007)	(0.011)	(0.010)
TRD	0.012 ***	0.015 ***	0.011 ***	0.016 ***
	(0.002)	(0.002)	(0.002)	(0.002)
FDI	−0.007 ***	−0.009 **	−0.008 ***	−0.004 *
	(0.002)	(0.004)	(0.003)	(0.003)
FOR	0.032 ***	0.014 ***	0.023 ***	0.028 ***
	(0.007)	(0.005)	(0.007)	(0.007)
PRG	0.327 ***	0.147 ***	0.287 ***	0.371 ***
	(0.061)	(0.045)	(0.071)	(0.054)
Constant	−3.156 ***	−1.096	−1.973 ^*^	−1.947 *
	(1.137)	(0.769)	(1.171)	(1.003)
Year fixed	Yes	Yes	Yes	Yes
Region fixed	Yes	Yes	Yes	Yes
*R*^2^/Pseudo *R*^2^	0.266	0.195	0.226	0.380
Obs	855	900	765	860

Note: *** *p* < 0.01, ** *p* < 0.05, and * *p* < 0.1; standard errors in parentheses.

**Table 6 healthcare-10-00030-t006:** Heterogeneity analysis results.

Variable	(1)	(2)	(3)	(4)	(5)	(6)
	NonOECD	OECD	Pre-Aged	Aged	Normal	Longevity
Burden	−0.235 ***	−0.147 ***	0.461	−0.214 ***	0.020	−0.212 ***
	(0.041)	(0.045)	(0.451)	(0.029)	(0.036)	(0.048)
URB	−0.059 **	0.132 ***	0.463	0.090 ***	0.092 ***	0.073 ***
	(0.027)	(0.016)	(0.415)	(0.012)	(0.020)	(0.023)
TRD	0.058 ***	0.010 ***	0.176 ^**^	0.008 ***	0.001	0.009 ***
	(0.007)	(0.003)	(0.085)	(0.003)	(0.005)	(0.003)
FDI	−0.002	−0.023	0.206	−0.005	−0.003	−0.005
	(0.003)	(0.016)	(0.263)	(0.005)	(0.006)	(0.007)
FOR	0.251 ***	0.048 ***	0.066	0.047 ***	0.028 **	0.054 ***
	(0.026)	(0.010)	(0.282)	(0.009)	(0.014)	(0.012)
PRG	0.421 ***	0.162 **	0.793 ***	0.226 ***	0.369 ***	0.331 ***
	(0.067)	(0.068)	(0.243)	(0.055)	(0.058)	(0.104)
Constant	−8.240 ***	−7.369 ***	−38.844 **	−0.910	−5.014 ***	0.322
	(2.147)	(2.236)	(15.567)	(1.438)	(1.834)	(2.858)
Year fixed	Yes	Yes	Yes	Yes	Yes	Yes
Region fixed	Yes	Yes	Yes	Yes	Yes	Yes
*R* ^2^	0.598	0.287	0.677	0.295	0.360	0.286
Obs	240	660	80	820	460	440

Note: *** *p* < 0.01 and ** *p* < 0.05; standard errors in parentheses.

**Table 7 healthcare-10-00030-t007:** Mediating effects test results.

Variable	Funds Resources		Variable	Labor Resources		Variable	Facility Resources	
	(1)	(2)		(3)	(4)		(5)	(6)
	THE	Balance		PHY	Balance		BED	Balance
Burden	0.092 ***	−0.058 **	Burden	0.023 ***	−0.100 ***	Burden	0.103 ***	−0.088 ***
	(0.008)	(0.024)		(0.006)	(0.025)		(0.019)	(0.025)
THE		−0.599 ***	PHY		−0.601 ***	BED		−0.244 ***
		(0.088)			(0.148)			(0.058)
URB	0.050 ***	0.099 ***	URB	0.007 **	0.073 ***	URB	0.036 ***	0.078 ***
	(0.004)	(0.012)		(0.003)	(0.011)		(0.007)	(0.010)
TRD	−0.007 ***	0.007 ***	TRD	−0.002 ***	0.010 ***	TRD	0.001	0.011 ***
	(0.001)	(0.002)		(0.001)	(0.002)		(0.001)	(0.002)
FDI	0.000	−0.006 **	FDI	−0.003 *	−0.008 ***	FDI	0.004 **	−0.005 *
	(0.002)	(0.003)		(0.002)	(0.003)		(0.002)	(0.003)
FOR	−0.013 ***	0.024 ***	FOR	−0.004 *	0.030 ***	FOR	0.046 ***	0.043 ***
	(0.003)	(0.007)		(0.002)	(0.007)		(0.006)	(0.007)
PRG	−0.164 ***	0.204 ***	PRG	−0.003	0.300 ***	PRG	0.138 ***	0.336 ***
	(0.025)	(0.058)		(0.010)	(0.060)		(0.030)	(0.062)
Constant	0.400	−2.262 *	Constant	0.205	−2.378 **	Constant	−2.836 ***	−3.194 ***
	(0.413)	(1.198)		(0.277)	(1.174)		(0.918)	(1.157)
Year fixed	Yes	Yes	Year fixed	Yes	Yes	Year fixed	Yes	Yes
Region fixed	Yes	Yes	Region fixed	Yes	Yes	Region fixed	Yes	Yes
*R* ^2^	0.650	0.289	*R* ^2^	0.582	0.268	*R* ^2^	0.458	0.267
Obs	900	900	Obs	900	900	Obs	900	900

Note: *** *p* < 0.01, ** *p* < 0.05, and * *p* < 0.1; standard errors in parentheses.

## Data Availability

Publicly available datasets were analyzed in this study and are attainable from the corresponding author upon reasonable request.

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
