# Peer review of "A Country-Level Empirical Study on the Fiscal Effect of Elderly Population Health: The Mediating Role of Healthcare Resources"

_healthcare, 2021, doi:10.3390/healthcare10010030_

Round 1
Reviewer 1 Report
Currently, many countries struggle with the problems of aging populations. The study examines the impact of the quality of health of the elderly on public finances. An interesting conclusion from the conducted research is the demonstration, based on quantitative methods, of the relationship between the health burden of the elderly and the aging of the population, and the impact of this process on public finances. In the opinion of the reviewer, the work is ready for publication.
Unfortunately, I have a problem with the analysis of the values ​​in tables 3 and 4. The paper does not describe what the numerical values ​​in parentheses mean. It is customary to give the values ​​of standard deviations in parentheses under the estimated values ​​of the variables. These measures cannot be negative because they are roots of variance. Moreover, the p-values ​​in these tables are surprising to me. I would like to see the data on which the calculations are made. If possible, I would like to recreate all the calculations in R. Unfortunately, in a short time I am not able to perform these calculations myself without access to the data.
Author Response
Point 1: Currently, many countries struggle with the problems of aging populations. The study examines the impact of the quality of health of the elderly on public finances. An interesting conclusion from the conducted research is the demonstration, based on quantitative methods, of the relationship between the health burden of the elderly and the aging of the population, and the impact of this process on public finances. In the opinion of the reviewer, the work is ready for publication.
Response 1: Thanks for your encouragement. The issue of population aging has been a hot topic in the world. Although many countries have stepped into aging society for a long time, there are still no one-size-fits-all measures or laws to follow. We still need to discuss and study the issue more deeply and extensively for better policies.
Point 2: Unfortunately, I have a problem with the analysis of the values in tables 3 and 4. The paper does not describe what the numerical values in parentheses mean. It is customary to give the values of standard deviations in parentheses under the estimated values of the variables. These measures cannot be negative because they are roots of variance. Moreover, the p-values in these tables are surprising to me. I would like to see the data on which the calculations are made. If possible, I would like to recreate all the calculations in R. Unfortunately, in a short time I am not able to perform these calculations myself without access to the data.
Response 2: In addition, we had reported the t-values in parentheses under the estimated values of the variables in the previous version of our paper. Anyway, to avoid misunderstandings, we have revised to report the standard errors in parentheses instead, and we have stated this at the bottom of each table in the latest version of our paper. And we used the software tool Stata 16.0 for all the statistic analysis.
Reviewer 2 Report
I think that the paper could be of interest, but I cannot understand its main contribution. I think the authors should try to clearly answer to the question: which is the novelty of this paper? Where is my contribution to existing literature? And they should start from this. It seems to me that the core of the paper lies in the mediation analysis. Nevertheless, comparisons with existing literature on this point is lacking.
In what follows, detailed comments for each section.
Introduction.
I would like an introduction where the topic relevance is stressed, not the statistics about the investigated phenomenon. Reduce general statistics and give more comparisons with previous studies on the same topics, given that no literature review is in the paper.
Which is the main problem you can help to address? In lines 43-47 you address main issues approached by literature. I would expect a clear statement about your paper compared to those issues.
Check sentence in lines 60-62.
The paper description is very confusing (73-76). I cannot find any literature review, nor correspondence among sections there described. This is a bad indicator of your work accuracy
Theoretical analysis.
The theoretical analysis is not strictly linked to your empirical analysis, given that there is no role for mediating effects. Furthermore – and this is fundamental – there is no comparison or comment concerning the whole literature framework on theoretical models about the issue. I would drop this section.
Empirical sections.
I find a potential endogeneity issue in your analysis. You should carefully address thi point with adequate tests. The result that ‘a rising elderly health burden leads to an increase in healthcare resources, which in turn has a negative impact on the fiscal balance’ is quite obvious and potentially endogenous. Furthermore, it is not linked to your conclusions. How can prevention health expenditure reduce (instead of increase) fiscal balance? Have you analysed a disaggregated measure of healthcare resources in terms of preventive/ cure expenditures? It seems to me that this point is not under your investigation, so you cannot argument on it.
Which kind of countries are under investigation? Is your study only about advanced countries? I cannot find this information that would be very relevant for your analysis. Institutional frameworks and health policies may have a relevant role in investigating this issue, so as development stage of each country. Better test countries’ differences to find interesting insights. You can add institutional indicators to derive soundness implications.
Conclusions
My suggestion is to limit conclusions to obtained results. You can add salience to your paper by adding some further refinements to the different impact of three main variable employed in the mediating variable (fund, labor and facilities). Can you find a different impact for these variables? Can you further empirically test their relevance? More interesting results could arise.
Author Response
Point 1: I think that the paper could be of interest, but I cannot understand its main contribution. I think the authors should try to clearly answer to the question: which is the novelty of this paper? Where is my contribution to existing literature? And they should start from this. It seems to me that the core of the paper lies in the mediation analysis. Nevertheless, comparisons with existing literature on this point is lacking.
Response 1: Thanks for your kind reminders. We have emphasized the main lines and highlighted the purpose of our study in the latest version of our paper.
In summary, we believe our contributions are as follows: (1) unlike previous studies focusing on the quantitative proportion of the elderly population, we empirically tested the health quality of the elderly population, which is a new perspective enlightening more attention on population quality; (2) we have proposed and estimated the conceptual framework of elderly population health, healthcare resources, and public finance, which can be regarded as a theoretical guideline for future research; (3) the indicator of the elderly health burden is a novel measurement of population aging, which can be used as a proxy variable in subsequent studies.
Point 2: Introduction.
I would like an introduction where the topic relevance is stressed, not the statistics about the investigated phenomenon. Reduce general statistics and give more comparisons with previous studies on the same topics, given that no literature review is in the paper.
Which is the main problem you can help to address? In lines 43-47 you address main issues approached by literature. I would expect a clear statement about your paper compared to those issues.
Check sentence in lines 60-62.
The paper description is very confusing (73-76). I cannot find any literature review, nor correspondence among sections there described. This is a bad indicator of your work accuracy
Response 2: Thanks for your comment. We have rewritten the Introduction section. First of all, we have enriched the literature review and summarized the literature from population aging and population health, respectively. [Lines 46-82]
Then, we have found less attention to the fiscal effects of the elderly health at the macroeconomic level. This is a research gap in the literature and a lack of empirical evidence. So we propose our conjecture that the determinant of fiscal deficit is the disease burden of the elderly population. [Lines 83-90]
Point 3: Theoretical analysis.
The theoretical analysis is not strictly linked to your empirical analysis, given that there is no role for mediating effects. Furthermore – and this is fundamental – there is no comparison or comment concerning the whole literature framework on theoretical models about the issue. I would drop this section.
Response 3: Thanks for your kind reminders. We have replaced “Theoretical analysis” with “Theoretical framework,” which includes two parts of “Section 2.1- Basic Model” and “Section 2.2 Mechanism Analysis”. [Lines 97-162]
After serious consideration, we still keep the theoretical derivation of the simple model because there is no relevant literature researching the relationship between disease burden of the elderly population and fiscal balance as far as we know, which is truly the primary relationship of our paper. We need it as the basic hypotheses for the later empirical part. [Lines 97-135]
In addition to enhancement the theoretical analysis that you pointed out, we have added some content of mechanism analysis based on the previous literature. And we have argued the mechanisms and channels of the elderly health burden and fiscal effects based on experience and literature. Finally, we have drawn a theoretical framework chart to demonstrate out clearly in Figure 1. [Lines 136-162]
Point 4: Empirical sections.
I find a potential endogeneity issue in your analysis. You should carefully address thi point with adequate tests. The result that ‘a rising elderly health burden leads to an increase in healthcare resources, which in turn has a negative impact on the fiscal balance’ is quite obvious and potentially endogenous.
Response 4: Thank you very much for pointing this out. We have added the “Section 4.2-Endogeneity Concerns” part to the paper. [Lines 264-283]
Due to the possible endogeneity problem of the model, we have used the lagged terms as the instrumental variable and used the system GMM model for further dynamic regression analysis. According to the literature, the system GMM approach is one frequently-used solution to the endogeneity problem such as simultaneity, variable omitted, which combines in a system, a regression in differences with regression in levels with the aim that additional moment conditions would be generated increasing the efficiency of resulting estimators.
In addition, to avoid measurement error, we replaced the variable measurements to re-run the GMM regressions by taking the ratio of net borrowing and lending to GDP (Finance) and the old-age dependency ratio (Aging) as an alternative.
As shown in AB(2) test and Hansen test, model specification is suitable, and the conclusions remain consistent.
Point 5: Have you analyzed a disaggregated measure of healthcare resources in terms of preventive/ cure expenditures? It seems to me that this point is not under your investigation, so you cannot argue on it.
Response 5: Thank you for your comments and suggestions. The analysis of the disaggregated measure of healthcare resources is a great idea. At the beginning of this research, we had thought about detailed analyzing the healthcare resources, especially healthcare expenditures as you mentioned. We believe that the specific item of health expenditures contains a lot of information that would be useful to disaggregate and analyze. However, due to the data availability and the lack of relevant literature, we, unfortunately, did not go further on this issue. Therefore, we regard this issue as a limitation of our research and a gap expected to focus on in future research.
Point 6: Which kind of countries are under investigation? Is your study only about advanced countries? I cannot find this information that would be very relevant for your analysis.
Response 6: Thank you for your comments. Due to the limitations of data availability, our study sample countries consisted of 45 countries as follows: Australia, Austria, Belgium, Brazil, Bulgaria, Canada, Chile, Colombia, Croatia, Cyprus, Czech Republic, Denmark, Estonia, Finland, France, Germany, Greece, Hungary, Iceland, Indonesia, Ireland, Israel, Italy, Japan, Korea, Latvia, Lithuania, Luxembourg, Malta, Mauritius, Netherlands, Norway, Poland, Portugal, Romania, Russian Federation, Slovak Republic, Slovenia, South Africa, Spain, Sweden, Switzerland, Thailand, United Kingdom, United States.
As shown above, they are not all advanced countries. Among them, there are 32 countries that are OECD members, and 12 countries are non-OECD members.
Point 7: Institutional frameworks and health policies may have a relevant role in investigating this issue, so as the development stage of each country. Better test countries’ differences to find interesting insights. You can add institutional indicators to derive soundness implications.
Response 7: Thank you for your nice suggestions. Considering country heterogeneity and following your instructions, we have further estimated the heterogeneous effects via group testing by country-level characteristics of economic, fiscal, and health conditions (such as income level, OECD member, elderly population proportion, budget types, life expectancy, healthy life expectancy, current health expenditures and so on). Due to maintaining content length and sample balance, we choose the representative results grouped by OECD member, aged society, and life expectancy presented in Section 4.4. Indeed, there are some interesting findings that the significant fiscal impact of the elderly health burden in aged society and longevity countries. We have tried to explain these findings according to the “cost of dying” mentioned in the literature, which performs higher healthcare spending as getting close to death. [Lines 310-336]
Point 8: Conclusions
My suggestion is to limit conclusions to obtained results. You can add salience to your paper by adding some further refinements to the different impact of three main variables employed in the mediating variable (fund, labor, and facilities). Can you find a different impact for these variables? Can you further empirically test their relevance? More interesting results could arise.
Response 8: Thanks for your suggestions. We have rewritten the Conclusion section. We focus on presenting our findings that the health burden of the elderly population has a negative fiscal impact on fiscal balance, and the consumption of health resources is one significant channel of mechanisms. Therefore, our final policy recommendations are proposed for how to reduce the elderly health burden and improve the utilization efficiency of healthcare resources. [Lines 399-419]
As for the comparison of the mediating variable (fund, labor, and facilities), we think this is a great research idea and professional opinion, which is also a direction for future research that can be extended. However, considering the purpose of our study is providing and estimating the conceptual framework of the elderly health, healthcare resources, and public finance, we have not putted the focus on the detailed analysis of healthcare resources and hope to fill the gap in future research. [Lines 393-398]
Reviewer 3 Report
The article titled as “A Country-Level Empirical Study on the Fiscal Effect of Elderly Population Health: The Mediating Role of Healthcare Re- sources” is very interesting and informative. However, the author (s) must incorporate the following issues to improve the article.
- How this study is different from the extant literature? The author (s) is required to explicitly mention the contribution/novelty of this study.
- The section 2.1 heading should be replaced as “Theoretical framework” instead of theoretical analysis and should be built with prior proper referencing from extant literature.
- One of the key issue with static models is that it suffers from endogeneity and omitted variable bias. Further, cross sectional dependency might also be an issue. How the results can be reliable in presence of these issues?
- Discussion of main results with contextualization i.e. consistency or contradiction with prior studies is very short. The author (s) needs to expand it.
- The policy implications have not been drawn from the main results. If possible, add it.
Author Response
Point 1: How is this study different from the extant literature? The author (s) is required to explicitly mention the contribution/novelty of this study.
Response 1: Thanks for your kind encouragement and reminders. We have emphasized the main lines and highlighted the purpose of our study in the latest version of our paper. In summary, we believe our contributions are as follows: (1) unlike previous studies focusing on the quantitative proportion of the elderly population, we empirically tested the health quality of the elderly population, which is a new perspective enlightening more attention on population quality; (2) we have proposed and estimated the conceptual framework of elderly population health, healthcare resources, and public finance, which can be regarded as a theoretical guideline for future research; (3) the indicator of the elderly health burden is a novel measurement of population aging, which can be used as a proxy variable in subsequent studies.
Point 2: The section 2.1 heading should be replaced as “Theoretical framework” instead of theoretical analysis and should be built with prior proper referencing from the extant literature.
Response 2: Thanks for your kind reminders. We have replaced “Theoretical analysis” with “Theoretical framework,” which includes two parts of “Section 2.1- Basic Model” and “Section 2.2 Mechanism Analysis”. [Lines 97-162]
After serious consideration, we still keep the theoretical derivation of the simple model because there is no relevant literature researching the relationship between disease burden of the elderly population and fiscal balance as far as we know, which is truly the primary relationship of our paper. We need it as the basic hypotheses for the later empirical part. [Lines 97-135]
In addition to enhancement the theoretical analysis that you pointed out, we have added some content of mechanism analysis based on the previous literature. And we have argued the mechanisms and channels of the elderly health burden and fiscal effects based on experience and literature. Finally, we have drawn a theoretical framework chart to demonstrate out clearly in Figure 1. [Lines 136-162]
Point 3: One of the key issue with static models is that it suffers from endogeneity and omitted variable bias. Further, cross-sectional dependency might also be an issue. How the results can be reliable in presence of these issues?
Response 3: Thank you very much for pointing this out. We have added the “Section 4.2-Endogeneity Concerns” part to the paper. [Lines 264-283]
Due to the possible endogeneity problem of the model, we have used the lagged terms as the instrumental variable and used the system GMM model for further dynamic regression analysis. According to the literature, the system GMM approach is one frequently-used solution to the endogeneity problem such as simultaneity, variable omitted, which combines in a system, a regression in differences with regression in levels with the aim that additional moment conditions would be generated increasing the efficiency of resulting estimators.
In addition, to avoid measurement error, we replaced the variable measurements to re-run the GMM regressions by taking the ratio of net borrowing and lending to GDP (Finance) and the old-age dependency ratio (Aging) as an alternative.
As shown in AB(2) test and Hansen test, model specification is suitable, and the conclusions remain consistent.
Point 4: Discussion of main results with contextualization i.e. consistency or contradiction with prior studies is very short. The author (s) needs to expand it.
Response 4: Thanks for your suggestions. We briefly have responded and compared the previous study with our findings in the Discussions section. In fact, we did not find the similar literature including the same variables and exploring the mediating effect as our study, but we have found some literature supporting the segmented relationship that population aging and health burden lead to increased consumption of health resources, and the health resource consumption affects government fiscal revenues and expenditures. [Lines 370-398]
Point 5: The policy implications have not been drawn from the main results. If possible, add it.
Response 5: Thanks for your suggestions. We have rewritten the Conclusion section. We focus on presenting our findings that the health burden of the elderly population has a negative fiscal impact on fiscal balance, and the consumption of health resources is one significant channel of mechanisms. Therefore, our final policy recommendations are proposed for how to reduce the elderly health burden and improve the utilization efficiency of healthcare resources. [Lines 399-419]
Round 2
Reviewer 2 Report
I find the paper sufficiently improved.
Reviewer 3 Report
Well dome and the article is much improved now.